# Implementation of Telerehabilitation in an Early Supported Discharge Stroke Rehabilitation Program before and during COVID-19: An Exploration of Influencing Factors

Louis-Pierre Auger [1,2,3,*] , Emmanuelle Moreau [3], Odile Côté [3], Rosalba Guerrera [3], Annie Rochette [1,2,3] and Dahlia Kairy [1,2,3]

1 School of Rehabilitation, Faculty of Medicine, Université de Montréal, Montréal, QC H3N 1X7, Canada
2 Centre for Interdisciplinary Research in Rehabilitation of Greater Montreal (CRIR), Montréal, QC H3S 1M9, Canada
3 Institut Universitaire sur la Réadaptation en Déficience Physique de Montréal, Centre Intégré Universitaire de Santé et Services Sociaux (CIUSSS) du Centre-Sud-de-l'Île-de-Montréal, Montréal, QC H3S 2J4, Canada
* Correspondence: louis-pierre.auger@umontreal.ca

**Abstract:** *Objective*: To identify the factors influencing the implementation of telerehabilitation (TR) in a post-stroke early supported discharge (ESD) rehabilitation program as perceived by clinicians and managers. *Methods*: A descriptive qualitative design was used in collaboration with a Canadian ESD stroke rehabilitation program. After 15 months of pre-COVID-19 implementation and 4 months of COVID-19 implementation, 9 stakeholders (7 clinicians, 1 coordinator and 1 manager) from an ESD program participated in 2 focus groups online or an individual interview. Qualitative data were coded and analyzed semi-deductively for the pre-COVID-19 and COVID-19 phases using the Consolidated Framework for Implementation Research (CFIR). *Results*: Four categories emerged related to the CFIR, each with themes: (1) Telerehabilitation, which included "Technology" and "Clinical activities"; (2) Telerehabilitation users, which included: "Clients' characteristics" and "Clinicians' characteristics"; (3) Society and healthcare system, which included "Changes related to COVID-19" and "ESD program"; and (4) TR implementation process, which included "Planning" and "Factors that influenced practice change". *Conclusions*: Factors impacting TR implementation in the ESD program were found to be numerous and varied according to the pre-COVID-19 or COVID-19 phases. Clinicians' motivation regarding potential gains for them in using TR was key in its implementation during the COVID-19 period.

**Keywords:** telerehabilitation; stroke; early supported discharge; integrated knowledge translation; implementation; COVID-19; Canada

## 1. Introduction

Stroke is a leading cause of disability worldwide [1]. An increasing number of people have a stroke each year and fewer die from it; therefore, more people presently live with the consequences of stroke [2]. Considering that stroke survivors are likely to need rehabilitation services to optimize function and social participation, and that the intensity and quality of services provided influence rehabilitation outcomes, innovative therapeutic options are needed [3].

Early supported discharge (ESD) programs offer intensive rehabilitation services to people who have had a stroke and are discharged home once medically stabilized. ESD services are believed to lead to better rehabilitation outcomes because of the intensity of care and home-based treatments that may allow better generalization of improvements in post-stroke individuals' daily life [4]. However, certain pragmatic factors, such as commuting between clients' homes and environmental uncertainties (e.g., snowstorms), may hinder clinicians' ability to offer treatments that meet the recommended intensity for

their clients [5,6]. Moreover, COVID-19 caused even more challenges to offering quality stroke rehabilitation services, including social distancing and the need to reduce contacts with clients, which brought to light the need to adapt usual practices in order to be able to provide care [7].

Telerehabilitation (TR), which consists of providing rehabilitation services from a distance using information and communication technologies [8], could be a promising tool for clinicians in ESD programs. TR has been shown to lead to significant improvement in functional outcomes and satisfaction, similar to in-person clinic-based rehabilitation post-stroke [9]. Recent reviews have shown that TR for stroke rehabilitation can lead to similar or better outcomes than conventional rehabilitation when used during the subacute and chronic stroke rehabilitation stages of care. A TR narrative review, led by Nikoleav and Nikoleav [10], included 70 studies and represented the examination of data from a total of 995 stroke survivors. TR was found to be feasible and to improve motor function (upper limb, lower limb and balance), cognition (spatial neglect, cognition and memory) and speech (aphasia). It was also shown to increase motivation and engagement, although technical issues and access to specialized equipment and the internet remained an obstacle [10]. Similarly, a recent Cochrane review of 22 randomized controlled trials of TR following stroke reported that outcomes between TR and conventional care were not significantly different [11]. Therefore, patients that receive TR during their stroke rehabilitation are likely to obtain similar results as compared to conventional in-person rehabilitation.

Hence, recent recommendations from the Canadian Stroke Best Practice Recommendations Virtual Stroke Rehabilitation Interim Consensus Statement 2022 [12] propose that TR for stroke rehabilitation should be provided across the continuum of care and considered as an alternative option to ensure access to and continuity of care. Few studies have examined the use of TR for early supported discharge from hospital. Laver et al. [11] included eight studies of TR post-hospital discharge; seven used phone calls to follow up and provide support to patients post-discharge and one used a combination of phone calls and emails. Van de Berg et al. provided a caregiver-supported exercise program for physical rehabilitation to supplement conventional stroke rehabilitation to be continued upon discharge with clinician follow-up using TR [13]. They reported a trend towards improved mobility and found it to be an acceptable and safe way of increasing exercise intensity at home. Thus, including TR as part of stroke rehabilitation is recommended and feasible, and leads to similar outcomes for post-stroke individuals. However, further studies are needed to better understand the role that TR can play for early supported discharge in order to facilitate TR implementation and adoption post-hospital discharge.

Furthermore, findings from previous studies were primarily obtained in structured studies that are not representative of the ecologic practice of stroke rehabilitation, such as randomized controlled trials [14,15]. To our knowledge, no study has focused on the implementation of stroke TR in the real clinical context of early supported discharge.

The objective of this study was to explore the factors influencing the implementation of TR in a post-stroke ESD rehabilitation program before and during COVID-19 as perceived by clinicians and managers.

## 2. Materials and Methods

### 2.1. Research Design

This study was conducted using a descriptive qualitative design [16], allowing the data analysis to be representative of participants' own experiences and perceptions on a little-known topic [17]. The qualitative study methodology used here is in line with the Consolidated Criteria for Reporting Qualitative Research-COREQ [18].

#### 2.1.1. Theoretical Framework

An integrated knowledge translation (IKT) approach was used throughout the study, from the study design until the end of the study [19,20]. By using an IKT approach, which

consists of involving stakeholders when conducting the study (e.g., implementation process, data analysis), the knowledge created is more likely to be useful and applicable by the stakeholders [21]. To do so, close contact was maintained between the research team and the ESD program, as the goal was to take an active part in the process to accompany the team during TR implementation. The Consolidated Framework for Implementation Research (CFIR, [22]) was used to guide the data collection and analysis. The CFIR [22] can be used to analyze the factors that influence implementation in five domains: the intervention (e.g., in relation to telerehabilitation), the inner (e.g., the clinical context where clinicians work) and outer (e.g., healthcare system, political context) settings, the individual characteristics (e.g., clients, clinicians, managers) and the implementation process (e.g., what is done, by who, when, in relation to implementation).

### 2.1.2. Population

The ESD program that collaborated with this study was part of a rehabilitation center in Montreal (Canada). This rehabilitation center offered services to people with various conditions, including stroke in the subacute and chronic phases. The ESD program offered treatment to individuals in the subacute phase of their stroke (i.e., up to three months post-stroke) [23], who persisted with mild-to-moderate deficits, that would have benefited from the same intensity of treatment as in an inpatient rehabilitation program and that were discharged home after receiving acute medical care. The ESD program's clients received home-based intensive rehabilitation (up to 5 days a week) for four weeks. The rehabilitation was tailored to clients' respective needs. The multidisciplinary team was composed of an occupational therapist, a physical therapist, speech language pathologist, a social worker, a clinical nurse and a special care counsellor. If needed, clients could also be referred to other disciplines such as nutrition and psychology. The composition of the ESD team in terms of the number of clinicians and variety of professional disciplines was representative of the two other ESD programs operating in the same city at the time of the study. For each client, an interdisciplinary treatment plan was prepared by the clinicians and the clinical coordinator based on the clients' respective needs, and weekly team follow-ups were held when needed. Upon discharge, clients stopped receiving rehabilitation services if they no longer needed them or may have been directed to other services, such as outpatient rehabilitation services or community-based services, according to their needs. The ESD program was implemented in the fall of 2017 and the current study started one year later. In fact, the implementation of TR was requested by the managers of the stroke rehabilitation program, for which the researchers were solicited and which led to the onset of the study. Preparations for TR implementation started in the summer of 2018 and TR implementation lasted until March 2020, at which time the hospital programs had to implement a number of significant sanitary restrictions related to the start of the COVID-19 pandemic. Telerehabilitation implementation then continued without intervention from the research team.

### 2.1.3. Telerehabilitation

Although TR can be performed by using various technologies, in this study, clinicians either used their work laptop or a computer workstation provided by the rehabilitation center with an external camera and microphone. During the pre-COVID-19 implementation period, the REACTS© telecommunication platform was used as it was one of the secure platforms accepted by the province's Ministry of Health. Zoom© was used during the COVID-19 implementation phase. On the clients' side, a personal cellphone or computer was used. During the pre-COVID-19 implementation phase, an iPad was provided to the ESD program for clinicians to lend it to clients, free of charge, when needed. For clinical activities relating to TR, clinicians were encouraged to apply clinical reasoning to their decision-making and could choose if and how TR would be used with their clients (e.g., fully on TR, or hybrid between in-person and TR).

### 2.1.4. Recruitment

The ESD program, and the coordinator and clinicians that were part of the program, were recruited by purposeful sampling [24]. Once the implementation of TR was completed, an email was sent to each clinician of the ESD program to suggest participation in the data collection of the study. To be included, potential participants had to have worked for at least one month in the ESD program during the implementation period, and they may or may not have used TR. The aim was to recruit as many members of the ESD team as possible to have a diversity of points of view.

### 2.1.5. Course of the Study

This study was conducted in three phases: (1) preparation, (2) pre-COVID-19 implementation and (3) implementation during COVID-19 (i.e., during the first wave of COVID-19). The third phase was added to the study because of the onset of COVID-19, which led to an opportunity to gather new and complementary experiences from participants to inform the feasibility of implementing TR.

The preparatory phase of the implementation lasted six months. During this phase, the implementation study was presented to the ESD clinicians, the technology (i.e., two iPad Pros, webcams and microphones) was ordered and training was provided to the clinicians and clinical coordinator (see Table 1 for detailed implementation strategies).

**Table 1.** Description and classification of the implementation strategies used during the pre-COVID-19 period (November 2018 to February 2020) according to the Consolidated Framework for Implementation Research [22] and Leeman et al.'s [25] classification.

| Strategies | Practical Application | Domains of the CFIR | Classes of Strategies |
|---|---|---|---|
| Training | A two-hour in-person training session was offered to the ESD clinicians and clinical coordinator regarding the use of the REACTS© platform. Introduction to the platform, chat and video conversations and document sharing were demonstrated and practiced by participants. Presentations, demonstrations and practical exercises were used. | Individuals (clinicians) | Capacity building |
| Transmission of professional guidelines regarding TR | Self-learning opportunities were sent to certain clinicians, such as the physical therapist. For example, certain professional orders (e.g., Quebec physiotherapy professional order) had created guidelines regarding the use of TR. These were transmitted to targeted clinicians. | Individuals (clinicians) | Dissemination strategy, capacity building |
| Periodic presence of the research coordinator at weekly team meetings | The research coordinator participated regularly (once to twice a month) in the ESD team's weekly meetings. The research coordinator observed the extent to which TR was discussed between the team and integrated in the services provided to clients. The research coordinator's presence also gave the opportunity for the ESD team to raise issues related to the implementation process and for the research coordinator to find solutions (see "ongoing troubleshooting"). The research coordinator's presence also acted as a reminder of the research project for the ESD team. | Process, individuals (clinicians) | Implementation process and integration strategies |

**Table 1.** *Cont.*

| Strategies | Practical Application | Domains of the CFIR | Classes of Strategies |
|---|---|---|---|
| Integration of a TR-related question in the interdisciplinary intervention plan canvas | The question "Would TR be relevant for this client, and if not, why?" was added to the standardized document used by the ESD team to prepare the interdisciplinary intervention plan for each client. The question was raised by the clinical coordinator at the time of the meeting, and clinicians discussed the relevance of using TR or not with each client. | Inner context, individuals (coordinator) | Implementation process |
| Ongoing troubleshooting | With frequent contact by email between the research coordinator and the ESD team, implementation issues were rapidly communicated and solutions were established collaboratively. Examples of troubleshooting were to resolve technical problems with the technology, putting clinicians in relation with technical support when needed, or ordering new material when needed. | Process, inner context | Implementation process, distributing messages |
| TR guides and protocols for use of technology | Guides and protocols were created in collaboration with technology specialists from the organization to (1) orient the use of REACTS© in practice for clinicians, (2) facilitate teaching to clients regarding using the technology (e.g., using the iPad Pro and connecting to REACTS©) and (3) ease problem-solving when technical issues arise (e.g., who to call, which number). | Intervention | Developing and distributing messages and materials |
| Establishment of a TR procedure and environmental configuration for clinicians | Specifically for when clinicians used TR to conduct team meetings (in the rehabilitation center) with clients (using TR in their home), guidelines regarding the configuration of the team meeting room, the material to use and how clinicians should be positioned to be visible by the clients via TR were established via a checklist and photographic descriptions. | Intervention, inner context | Developing and distributing messages and materials |
| Pretesting technology before an attempt | For the majority of TR intervention, a pretest of the technology was realized between the clinician and a member of the research team. During these pretests, the quality of the internet connection, the functioning of the camera and microphone and the ability of the clinician to properly connect to the platform were assessed. The pretest was conclusive when TR technology was found to be functioning as intended. | Intervention | Integration strategies |
| Regular contacts between research and ESD teams | Regular emails and live discussions were exchanged between the research and clinical coordinators, and when needed with clinicians, regarding the project. These contacts ranged from several times a week to once a month depending on the project phase and the needs of the ESD team. | Process | Integration strategies |
| Bi-annual meetings between research and ESD managers | Meetings were organized between the research team and the coordinator and managers of the ESD program in order to discuss the implementation project and the main barriers and facilitators that were experienced, and to plan the interventions that needed to be realized in order for the project to function optimally. | Process | Implementation process strategies |
| Adapted support for TR sessions | At the request of the clinical team, a member of the research team was available to be present in preparation, during and after the TR session (either physically or virtually) to support clinicians. | Intervention, individuals (clinicians) | Integration strategies |

The pre-COVID-19 implementation phase lasted 15 months (until February 2020). During this phase, the ESD clinicians and coordinator continued their usual practice and were encouraged to use TR with their clients when deemed relevant. To foster implementation, various strategies were used by the research team. These are described in Table 1 and categorized according to the CFIR domains [22] and the classification of implementation strategies by Leeman et al. [25].

The implementation phase started in March 2020 and lasted until the end of data collection in July 2020. During that phase, no intervention from the research team occurred, since all formal research activities were put on hold due to hospital policies related to the pandemic, and the ESD program was free to use TR as they deemed relevant.

### 2.1.6. Data Collection

#### Sociodemographic Characteristics

Age, gender, professional discipline and the number of years of experience in stroke rehabilitation were collected for each participant.

#### Factors Perceived as Influencing Implementation

The implementation process was documented from its beginning to its end in a log by the first author, where events related to the implementation study (e.g., training, TR session), discussions with members of the ESD or the research team, and his own perceptions as the research coordinator were written. Moreover, a semi-structured interview was conducted in August 2019 with a clinician that was going on maternity leave. Finally, two focus groups were conducted at the end of the COVID-19 implementation period: one for clinicians and one for the coordinator and the manager. Each focus group was semi-structured and co-animated by the first and last authors. For each focus group, facilitators used an interview guide that was based on the CFIR [22], the expertise of the last author in knowledge translation and implementation science, and questions to compare the pre-COVID-19 and COVID-19 implementation periods. Data collection was realized remotely and recorded using the Zoom Pro platform to respect sanitary measures related to COVID-19. The focus groups' duration was between 75 and 90 min, and the interviews lasted around 60 min. The interview and focus groups were transcribed verbatim and uploaded to the QDA Miner software to facilitate organization and coding of data.

### 2.1.7. Data Analysis

Sociodemographic characteristics were analyzed using descriptive statistics. To preserve participants' confidentiality, the range of age and number of people per gender is presented instead of an individual description of each participant. The transcripts from the focus groups and interviews were coded by the second author using a coding scheme based on the CFIR [22]. In line with the semi-deductive thematic analysis approach that was used, additional codes and themes that emerged from the data were added to the coding scheme. Data related to the pre-COVID-19 and COVID-19 phases were highlighted when possible. Regular meetings were held between the coder and the research team to further enrich the thematic analysis and foster understanding of the themes extracted. Themes were categorized according to the CFIR domains [22] with related themes and sub-themes. Data from pre-COVID-19 and COVID-19 implementation phases were analyzed individually and then compared and contrasted to identify similarities and differences in order to better understand factors underpinning the uptake of TR. The thematic analysis was shared with participants during an ESD program meeting and their feedback was collected to validate and deepen analysis.

The first and last authors were the main investigators involved in the data collection and analysis. From an ontological perspective, their position was that TR could be a potentially useful and relevant tool to integrate in stroke rehabilitation, including in the early supported discharge program. They are both rehabilitation clinicians with clinical experience. The request to integrate TR into the ESD program came from the clinical

team and the research team's role was to provide the necessary support and resources. Ongoing reflections regarding the research team's role in the implementation process were documented in the implementation log. From an epistemological perspective, the authors adhered to a constructivist perspective of knowledge, according to which the relevant knowledge to be studied and highlighted in this article was that which came from the combination of the different stakeholders involved in the study (participants and researchers) [26]. The authors were also influenced by the integrated knowledge translation paradigm throughout the study, from its design until data analysis.

2.1.8. Ethical Considerations

This study was approved by the Research Ethics Board of the Centre for Interdisciplinary Research in Rehabilitation of Greater Montreal (CRIR). Each participant signed an informed consent form before taking part in the study and was free to withdraw at any time.

**3. Results**

*3.1. Description of the Sample*

A total of nine stakeholders from the ESD program participated in the study, and only one person, the nurse, did not respond to the invitation to participate in the study for unknown reasons. The majority (*n* = 7) of participants were of female gender and the range of age was between 29 and 59 years old. Participants' ID and professional disciplines are described in Table 2.

**Table 2.** Participants' professional characteristics.

| Participant ID | Discipline | Years of Experience in Stroke Rehabilitation |
|:---:|:---:|:---:|
| 1 | Occupational therapist | 1 |
| 2 | Occupational therapist | 5 |
| 3 | Special care counsellor | 3 |
| 4 | Social worker | 2 |
| 5 | Social worker | 10 |
| 6 | Speech language pathologist | 13 |
| 7 | Physical therapist | 10 |
| 8 | Clinical coordinator | 15 |
| 9 | Manager | 36 |

*3.2. Factors Perceived as Influencing Telerehabilitation Implementation*

Four categories based on the domains of the CFIR [22] were extracted from the thematic analysis of data: (1) telerehabilitation, (2) telerehabilitation users, (3) healthcare system and society and (4) implementation process. The categories with related themes and sub-themes (if applicable) are summarized in Table 3 and presented in detail below.

**Table 3.** Perceived factors influencing the implementation of TR in the ESD program.

| Categories with CFIR Domain | Themes | Sub-Themes |
|---|---|---|
| I. Telerehabilitation (CFIR domain: Intervention) | Technology | Internet connection |
| | | Platform |
| | | Equipment |
| | Clinical activities | Assessment |
| | | Intervention |
| II. Telerehabilitation users (CFIR domain: Individual characteristics) | Clients' characteristics | Socio-demographic characteristics |
| | | Post-stroke condition |
| | | Availability of a caregiver |
| | Clinicians' characteristics | Therapeutic relationship |
| | | Beliefs and attitudes |
| | | Scope of practice |
| III. Society and healthcare system (CFIR domain: inner and outer stings) | Changes related to COVID-19 | Impact on perceptions towards TR benefits |
| | | Impact on ESD services' expectations |
| | | The ESD as a buffer for the stroke rehabilitation continuum |
| | The ESD program | Nature of the service |
| | | Interdisciplinary teamwork |
| IV. TR Implementation process (CFIR domain: implementation process) | Planning | Collaborating with researchers |
| | | Telerehabilitation adaptation methods |
| | Factors influencing change of practice | Staff mobility |
| | | Champions |
| | | Innovation culture |

CFIR: Consolidated Framework for Implementation Research [22]; ESD: early supported discharge.

I. Telerehabilitation

With regard to TR as it was used by the ESD, two main themes related to "Technology" and "Clinical activities" emerged.

Technology

Internet connection*:* The stability of the internet connection influenced the quality and fluidity of the transmission of sound and visual information during TR sessions. However, technical problems related to the internet connection did not constitute a barrier to TR implementation, users generally having a sufficiently good connection to hold a TR session. "[ . . . ] *once, a lady* [for whom] *the Wi-Fi was so-so, but maybe it was the connection in the rehabilitation center with which the clinician had more difficulty, but it was maybe one out of twenty* [1/20] *clients that we had.*" (P8—Clinical coordinator)

Platform. The login to the telecommunication platform for the client was an important aspect for the clinicians. Pre-COVID-19, clinicians used the REACTS© platform, which at the time required that an account be created on the clinicians' side, and that clients had to log in using an email address and a password. These procedures led to difficulties on both ends of the platform. The REACTS© platform suggested to its users many functionalities,

such as videoconference, screen sharing, document sharing and other functions, although clinicians only used videoconferences at the time. At the time of the outbreak of COVID-19, the health establishment allowed clinical programs to use widely commercially available platforms which were increasingly used for telehealth at that time for follow-ups, such as Zoom: *"The fact that it was Zoom, it was extremely easy. [ . . . ] You click on an internet link, [ . . . ] then you can use Zoom."* (P8—Clinical coordinator). In fact, clinicians reported greater ease of using Zoom for their clients to connect to TR sessions. In addition, Zoom met their clinical needs, considering that most only used video conferencing and screen-sharing functions in their follow-ups. Thus, in this clinical context, the ease of use of the platform appeared to be more important in the eyes of the stakeholders compared to the range of functions that the platform (e.g., instant sharing of files) could offer them.

Equipment*:* Pre-COVID-19, the ESD program used an iPad which was loaned to users to carry out TR. The clinicians had to integrate the evaluation of the ability to use the iPad into their clinical follow-up and, if necessary, teach its use to the user or their family members. This generally constituted a barrier to its integration into pre-COVID-19 clinical monitoring. However, the iPad was generally well accepted by the few users who received it. During the implementation period which took place during COVID-19, the electronic equipment used was the clients' own in order to limit the spread of infection. This impacted on the ability to offer TR services. Indeed, depending on the clinical activities to be carried out and the required point of view, the mobile nature of the equipment influenced the TR sessions. For example, in physiotherapy, the professional had to be creative to observe movements. *"[ . . . ] there are some* [clients] *with whom I just wasn't able to do the interventions I wanted due to their technological equipment, like the computer which could not be moved. On the other hand, there were people, with the tablet, for whom it went really well and they moved it around for me and they placed it a little bit anywhere and it was fine."* (P7—Physical therapist). The impact of the type of equipment on the service offered was perceived differently depending on whether it was an activity that could be carried out in front of the computer (e.g., paper and pencil tasks) or that required being able to observe the person interacting with their environment (e.g., walking in the hallway, preparing meals in the kitchen). *"[ . . . ] I did an artistic, fine motor activity with a client and I lowered my screen . . . you know . . . so that the camera could see what I was doing and we were doing the painting activity at the same time."* (P3). However, mobile equipment, such as cell phones, could sometimes be difficult to position once they had been moved by customers, considering for example the impairment of their upper limb motor skills or less familiarity with technology.

Clinical Activities

Assessment*:* Some clinicians perceived their TR assessment approach as less objective and specific as compared to when it was carried out in person. *"*[To] *assess an affected upper limb when the other person is in front of a screen [ . . . ] it is certain that our assessments have been much more subjective than objective. We are not* [physically] *there to measure the amplitude, we are not there to do the Jamar* [dynamometer], *[ . . . ] but we were in a pandemic."* (P8—Clinical coordinator). The fact of having the possibility of having a global vision of the person within their environment, when physically there, was also reported as a factor influencing the evaluation process: *"[ . . . ] it's true that when we go to people's homes, [ . . . ] there are loads, loads, loads of information [ . . . ] that lead us to ask lots of questions, to check all sorts of things that you can't do when you're just with them in front of a screen."* (P5—Social worker). Certain aspects of the assessment, particularly related to safety, such as dysphagia and risk of falling, could not generally be carried out via TR: *"When we were wondering about the safety of the patients' returning at their home, [ . . . ] I find that it really took a home visit to see if they were safe [ . . . ] It was security elements [ . . . ]."* (P1—Occupational therapist) and *"[ . . . ] in terms of dysphagia too, of course . . . there are assessments that I can do in TR, but you know if I have a client who is at risk of not swallowing food properly and suffocation, I prefer to be there."* (P6—Speech language pathologist). Therefore, regarding assessment using TR, the clinicians reported feeling generally limited in terms of the methods they could use, as well

as the scope of their observations, which was limited to what the camera transmitted to them.

Intervention: Adaptations were required in the practice of certain clinicians, such as adjusting the progression of exercise programs and the teaching given to relatives. This was influenced by the availability of equipment in the patients' homes, as well as the risk associated with the suggested exercises. Among the clinicians for whom TR had the greatest impact on the interventions was the physiotherapist: "*We realize that, well, it's clear that, for the more difficult exercises, I'm not necessarily going to do them unless there's, you know, there really has to be the caregivers really close and that they understand what can happen so, I talk, I explain a lot more what is going to happen in the exercise, what are the risks,* [ . . . ] *where to put their hands, etc.*" (P7—Physical therapist). Thus, professionals sometimes had to choose to reduce the complexity, variety or intensity of some of their interventions when they used TR.

## II. Telerehabilitation Users

With regard to telerehabilitation users, two main themes related to "Clients' characteristics" and "Clinicians' characteristics" emerged.

### Clients' Characteristics

Sociodemographic characteristics: Client age and socioeconomic status were noted by clinicians as playing a role in the decision to use TR. Thus, a clinician noted that the clientele seen at the CPA was generally older: "[ . . . ] *there is this notion where we have an aging clientele which is often 80 on the way up. The use of technologies is often not in their habits, in their daily life.*" (P4—Social worker). Thus, individual client characteristics may have an impact on the ease or approach clinicians take when using TR.

Post-stroke functioning: The degree of impact of the stroke on clients' functioning influenced the propensity of clinicians to use TR. In fact, the degree of cognitive, language and motor impairments were mentioned by clinicians as having an impact on the ability of clients to connect and use technology for a TR session: "*At some point, when you are already handling an iPad* [with cognitive and physical difficulties] *and you're all alone at home because you don't have anyone else,* [ . . . ] *it doesn't make much sense anymore* [to use TR]." (P4—Social worker). In addition, fatigue linked to the use of technology was raised as a limiting factor: "*The clients, . . . when there have been on two Zooms in their day . . . Let me tell you that they are not asking for a third.*" (P8—Clinical coordinator). Managing schedules for the client was also an issue reported by clinicians, particularly some sessions were carried out face-to-face and others via TR in the same day: "*So they* [Clients] *really have to have a cognitive level that allows them to organize the schedule to be able to understand* [if next therapy will be with TR or in-person]." (P2—Occupational therapist).

Availability of a caregiver: The presence and availability of a caregiver was perceived as facilitating by participants in the use of TR for clients. A clinician was concerned about the client's ability to use TR: "[ . . . ] *my fear is always: does the person have the capacity* [to use TR]. *If there is a relative, we are ok. If there isn't any, we are less sure.*" (P4—Social worker). The context of the pandemic had an influence on this aspect: "*We had a lady, among others, 85 years old, who lived with her son* [ . . . ]. *Well, he wasn't working. He was on leave and he was the one who set up all the Zoom meetings for his mother. Then when she was in front of the screen, he would leave.*" (P8—Clinical coordinator). Thus, the presence of a loved one, such as a child or partner, generally made it easier to manage the technology, allowing clients to connect and participate in the TR session.

### Clinicians' Characteristics

Therapeutic relationship: The clinicians mentioned having fears at first about not being able to develop a therapeutic relationship if follow-ups would be carried out by TR: "*The link is there, it's still there, it's exactly the same thing.*" (P7—Physical therapist). The therapeutic relationship which developed was found to be bidirectional: "[ . . . ]*You know, I really develop a form of therapeutic attachment. And it's the same thing on their side* [clients], *they*

*tell us even if we've never seen each other or been close, touched, we have the impression, they have the impression, that we are part of their life.*" (P6—Speech language pathologist). Thus, the experience of the clinicians suggests that a therapeutic relationship can be developed both during TR and in-person sessions.

Beliefs and attitudes: The use of TR by clinicians was influenced by their previous experiences, their sense of self-efficacy and their perception of the quality of the services they offered. A clinician reported a lack of interest in TR because of technical problems she had experienced pre-COVID-19 and her preference to carry out her follow-ups in person: "[ . . . ] *for me it was so energy-consuming to call and try to solve the* [technological] *problems that it was much faster to take my car and go to the customer.*" (P6—Speech language pathologist). With the context related to the pandemic, the risks of contamination associated with face-to-face follow-ups with clients led to a shift where stakeholders had to repeatedly use TR and thus feel more comfortable with this modality. "*Over time I allowed myself to do more and more* [TR], *and I find that COVID really pushed me a lot, which means that I find that I have . . . I use technology more with clients.*" (P6—Speech language pathologist). Thus, beliefs and attitudes regarding TR were positively influenced by repeated exposure to TR during the COVID-19 pandemic.

Field of practice: Concerns regarding the use of TR for clinical activities, and adaptations made, differed between clinicians of different disciplines as their fields of practice differed. Due to concerns about the quality of the services offered, as well as the safety of the clients, all therapists reported limitations in the use of TR and a preference to return to face-to-face services once the sanitary restrictions would allow it. A physiotherapist explained that with regard to managing fall risk: "*I want to make sure that there is no risk of falling and the exercises that I am going to do are generally exercises that are difficult for patients, therefore putting them at risk of falling* [ . . . ] *I'm going to be more effective when I'm in their home and doing my interventions.*" (P7—Physical therapist). One of the occupational therapists on the team mentioned concerns regarding the risk of burns during meal preparation and a preference to be on site with the client during the assessments and interventions. "*Because . . . when we have no choice* [to use TR] *we do it. If we go back to doing home visits, it's possible, I think we might go back a bit to . . . what we did in the past . . .* " (P2—Occupational therapist). Thus, the clinical activities related to the different professional roles led practitioners to more or less integrate TR into their practice depending on their professional disciplines. In general, physiotherapists and occupational therapists seemed more inclined to include face-to-face follow-ups, while speech therapists used remote follow-ups more often.

III. Society and the Health Care System

With regard to society and the health care system, two main themes related to "Changes related to COVID-19" and "The ESD" emerged.

Changes related to COVID-19

Impact on perceptions towards TR benefits: COVID-19 had an impact on clients and clinicians in terms of the acceptability of TR. Indeed, the fear related to the spread of COVID-19 contributed favorably to the adoption of TR for clients and clinicians: "*Patients were less and less interested in seeing us [in person] at the start of COVID. They were starting to be more and more worried.*" (P6—Speech language pathologist) and "*You know, we didn't know what the virus was and then a pandemic. So . . . yes there were fears among the clinicians at the start* [of the pandemic] *of going to their* [clients'] *home. So . . . they weren't against using Zoom.*" (P8—Clinical coordinator). In addition, the context of COVID-19 and the increased use of TR led clinicians to consider using TR once the pandemic would end: "*You know, I was thinking about the days when we had bad weather last winter and we just canceled everyone. If ever there are people who actually have the possibility of doing TR* [in the future], *well . . . maybe on those days we can already have that ready* [to use] *so that they don't lose a day or two of therapy.*" (P2—Occupational therapist). TR has therefore become, in the eyes of clinicians, a means of preserving their safety and their health, as well as that of their clients, in many different situations, including a pandemic.

Impact on ESD services' expectations: Because of COVID-19 and its multiple impacts on the health system, health services in Quebec were more difficult to access. For this reason, TR was more readily accepted by clients and clinicians, as it was a way to continue providing services despite social distancing guidelines. *"What was interesting with COVID was that the expectations were a little bit lower in terms of rehabilitation, in the sense that it's better than ... better than nothing.* [ ... ] *So, even if we perhaps did not address all the issues or not as deeply, well at least they* [the clients] *had a follow-up.* [ ... ]. *Then I think that the fact, you know, that the clients were kind of grateful to have therapies when all of Quebec was closed, they were perhaps a little more tolerant of difficulties that we could sometimes have with technology."* (P6—Speech language pathologist). Disruptions linked to COVID-19 therefore led to the idea of "Better than nothing" during the peak of the pandemic in the eyes of clinicians and clients, which allowed them to avoid service disruptions at the ESD.

The ESD as a buffer for the stroke rehabilitation continuum: The context related to COVID-19 put pressure on the Quebec health network and forced the ESD program to adapt the services it offered for stroke clients. *"First, there are some stroke patients who did not even show up at the hospital because they did not want to catch COVID. It was even worse when they were offered to go to inpatient rehabilitation and be in their room for fourteen days* [Preventive isolation COVID-19 precautions upon admission], *well, that didn't interest them either. So they refused inpatient rehabilitation. For outpatient rehabilitation, we had to limit it to essential services and transfer most outpatient clinicians to help in long term care. So the outpatient services provided were at a minimum. So it was really the ESD that was affected from both sides. They wanted to be able to stay home so were referred to the ESD."* (P9—Manager). In fact, at the height of the first wave of COVID-19 in Quebec (spring 2020), the CPA practically doubled the number of clients served ($\approx$10 active follow-ups vs. $\approx$6 pre-COVID-19) by temporarily adapting their services offered (e.g., lowering frequency of treatment by certain clinicians whose discipline was less of a priority for the client). These adaptations in terms of the services offered also led to slight changes in the clientele treated. The stroke patients had less severe deficits, which was a factor facilitating the implementation of TR: *"It must also be said that our clientele* [ ... ] *was older before COVID than during COVID. Also they were less handpicked,* [in pre-COVID-19] [ ... ] *the coordinator made sure that there would be relatives present and who would provide support for TR."* (P6—Speech language pathologist). TR was therefore one of the means used by the ESD, with the adjustment of the client selection criteria, to serve more clients and limit service disruptions for post-stroke individuals who would typically have been referred to inpatient or outpatient stroke rehabilitation pre-COVID-19.

The ESD Program

Nature of the service: The context surrounding the CPA team, as well as its specific service offer (i.e., ambulatory), had an influence on the implementation of TR. Indeed, all services were offered at home pre-COVID-19, but clinicians already had the equipment required to telecommute due to their need for daily mobility. *"Our team, the fact that it is a dedicated and equipped team has helped in the implementation of TR. They were already teleworking, but only for administrative tasks. So their charts, things like that, statistics, they were already doing that at home* [Pre-COVID-19]." (P9—Manager). Thus, ESD clinicians were already familiar with the concept of telework for their administrative tasks (excluding interventions with their clients), which facilitated their adaptation when COVID-19 came.

Interdisciplinary teamwork: Organization within the interdisciplinary team facilitated the implementation of TR. More specifically, the communication mechanisms present within the team as well as the support offered between peers were well developed. *"Instructions were clear and precise. There was no ambiguity, so we went with that."* (TES). Communication was also helpful when team members encountered technical glitches: *"We found solutions together too. Instead of calling* [technical services], *well, we called each other, we texted each other, we tried to find the answer and we were more in a trial-and-error mode."* (P5—Social worker). Clinicians therefore favored solving their technical problems among themselves rather than asking for support from the technical services of their organization. In addition, in terms of peer support, the coordinator was of great help for all the clinicians on the team in terms of

choosing and adapting the clinical activities to be carried out, but this time when doing them using TR. "*They often called me to find out what I thought of it* [Intervention plans] [ . . . ] *Especially with cognitive disorders, how do you assess that* via *Zoom. In occupational therapy, they asked me questions about that. In specialized education too, there were a lot of questions like "how am I doing that?", "Can you help me?", "Does that make sense?"* (P8—Clinical coordinator). Between colleagues, there was an enhancement of interdisciplinary collaborative work through closer communication to optimize home visits, which the team aimed to keep to a minimum. "*I often asked the occupational therapist, who needs to go to clients' home more* [ . . . ] *to do a few little things that I couldn't do with the camera, so yes, that changed* [work with colleagues]." (P7—Physical therapist). Similarly, a social worker also mentioned having tried to reduce her travels and therefore collaborate more with other colleagues: "*I find that I relied a lot more on my colleagues for the evaluations, the objectives, the interventions that were made there, to see, was it necessary for me to be the one doing the talking to the caregivers, you know. All of that made me think a lot more* [about my ways of doing things]." (P5—social worker). Interdisciplinary work and close communication therefore allowed TR clinicians to take advantage of the presence of other clinicians going to clients' homes to complete their assessments or for interventions that could not be performed remotely (e.g., having a document signed).

IV: TR Implementation Process

With regard to TR implementation process, two main themes related to "Planning" and "Factors influencing change of practice" emerged.

Planning

Collaborating with researchers: The ESD program team had already been exposed to TR as part of this research project (i.e., pre-COVID-19 period): "*Well, I think the fact that we already had the project in place and the team was already made aware of that [TR], and we had been trying to find places to set it up for two years.*" (P9—Manager). The manager recognized that participating in the research project pre-COVID-19 facilitated their own implementation of TR with the clinical team at the time of the onset of COVID-19. In fact, no additional intervention was made by the research team during TR implementation which occurred after the onset of the COVID-19 pandemic.

Period of adaptation to TR: Following the decision to offer TR services, clinicians took time to think about how their practice could be adapted to TR: "*I think that for a day or two we prepared for each client, ensured that* [ . . . ] *the client could have the connection* [to the internet and to the Zoom platform]. *And gradually there was a transfer of* [clinicians] *who were first just going to patients' homes, and then that used only Zoom, and we would not go home even right after their* [clients] *return from the hospital.*" (P2—Occupational therapist). This occupational therapist further explained the approach she chose to adapt her practice to TR: "*So, at the start, I said to myself: how am I going to do occupational therapy remotely? I prepared a Google Docs by activity of daily living, by skills, and how I can look* [at these], *how I can evaluate* [these] [ . . . ]. *At the start it was really, bottom-up, you know . . . capacities or the basic things, and after that I said to myself, I can still do some observations, so for dressing, you still have to analyze the person. I also did some meal preparations.*" (P2—Occupational therapist). TR adaptation occurred gradually as it was used progressively. This required time from the clinicians, which did not immediately lead to improved productivity or increased number of clients they could see per day: "[it is] *A new way of doing our interventions, so it's sure that it requires more planning. then well as you say, the commuting is not necessary now but . . . also while commuting, we planned our therapies.*" (P6—Speech language pathologist). Thus, the potential benefits of using TR, such as reducing travel time to clients' homes, were not experienced by clinicians in the early stages, because adaptation to TR required more time and effort from them.

Factors Influencing Change of Practice

Staff mobility: The fact that there was turnover within the ESD team during the pre-COVID-19 period hindered the implementation process. For example, new clinicians on the team did not receive the training for TR that previous or senior team members had.

Champions: The structure of the team, including the presence of the clinical coordinator, constituted a facilitator in the process of change. Moreover, the ESD team was unanimous as to the importance of a resource person within the team who was comfortable with using technology. This person could explain how to use the technology and, subsequently, was a contact person for their colleagues. "*Well I think it definitely takes someone from the team who has more computer skills than me, in this* case [P2] who *was a really valuable help.*" (P5—Social worker) Depending on the implementation period, staff mobility varied. The role of champion was generally adopted spontaneously by the occupational therapist in the team.

Innovation culture: The ESD team was able to adapt to TR quickly and find diversified strategies to maintain an efficient service, which was associated by participants with the innovation culture in the team. "*Of course the ESD team is a team that is used to driving there* [clients' homes], *you know . . . they are efficient, and it's a team that adapts very well to change.*" (P8—Clinical coordinator). An occupational therapist from the team also recognized the speed with which each of the members was able to adapt to the situation: "*Because it takes a lot to make you change your practice* [ . . . ]. *I find that we made the turnover in around 24 to 48 h, really . . .* " (P3—Occupational therapist). Clinicians adapted their interventions and found creative ways to meet the needs of clients using TR, either using reference tools (e.g., individualized videos and information capsules) or adapting the services offered (e.g., integrating group therapies). This allowed services to be offered to clients that would not have had access to services without TR; for example, due to environmental factors (e.g., snowstorms, infestations) that could have affected clinicians' or clients' safety. More specifically, regarding educational tools, the speech therapist and the special care counsellor of the team adapted the content of the information transmitted to the customers: "*So for example, I do my programs using videos. I film myself doing it, then I send it to the client. Something I didn't do before. I used paper and pencils.*" (P6—Speech language pathologist). By carrying out the interventions online, the clinician also realized that it could be interesting to rethink the services offered for certain interventions: "*So, we do them* [Exercises], *although sometimes it can become a little boring. So, I do them at the same time as them* [Clients], *you know it's like a class,* (laughs) *Zumba . . . We try to make it a little bit festive.*" (P3—Special care counsellor). Clinicians were therefore proactive not only in replicating the intervention methods they already used pre-COVID-19, but also used technology to rethink certain intervention methods, to make them more attractive or optimize their working methods.

## 4. Discussion

This study explored the factors influencing the implementation of TR in a post-stroke ESD rehabilitation program according to clinicians and managers. To our knowledge, this is one of the first studies exploring TR implementation and use in a real-life setting with an interdisciplinary team and which occurred partially during COVID-19, as opposed to other research dedicated to the impact of specific technologies or treatments on post-stroke individuals' recovery outcomes. This study suggests that multiple factors related to TR, clinicians and clients' characteristics, the healthcare system, the ESD program and the implementation process influenced the implementation of TR. The comparison of pre-and per-COVID-19 periods further helped explain the role that these factors played in TR implementation.

In our study, COVID-19 was found to have acted as a catalyst for the implementation of TR, as has recently been shown [27]. Moreover, the main influencing factors that had an impact on the implementation of TR were highlighted, as well as their variation over time and according to the context. For example, pre-COVID-19, the platform used and the limited experience with it was a barrier to TR implementation, while the use of an increasingly used platform during the COVID implementation period was a facilitator.

Moreover, pre-COVID-19, ESD clinicians had little incentive to use TR due to a perception that the burden of changing practice did not outweigh the benefits for their clients. This was, among other things, related to the technology being more complex than what was required for their needs at the time, as well as the perception that face-to-face follow-ups should be offered to offer optimal quality of service. Similarly, the need to understand the usefulness and the preference for face-to-face follow-ups also emerged as important influencing factors in a review of six Canadian studies of the implementation of TR in post-stroke rehabilitation in Canada [14], as well as a recent multiple case study in the field of TR and stroke [28]. Moreover, knowing that outcomes and resources used for in-person and TR follow-ups can be similar, it is not surprising that clinicians lacked motivation in the pre-COVID-19 phase of our study to make the effort to change their practice [14]. The low uptake of TR prior to the onset of the COVID-19 pandemic was similarly reported in a recent study showing that few physiotherapists used TR prior to COVID-19 [7]. Finally, challenges related to the technology used has been also recognized as a barrier to TR implementation [29].

When COVID-19 sanitary measures were put into place in most places in March 2020, the health instructions from the ESD program's organization recommended restricting face-to-face follow-ups as much as possible because of the fear of contamination between clients and clinicians. However, stakeholders in the ESD program were concerned about maintaining the best possible quality of service, which motivated the professionals to integrate TR into their practice. Caughlin et al. [14] also showed in their study that clinicians will opt for TR when in-person follow-ups are not possible, regardless of the reason. By following a gradual implementation process and supporting each other through self-appointed champions in their team, ESD clinicians and managers used certain strengths of their program, such as interdisciplinarity and being accustomed to performing administrative tasks remotely, to adjust their service offer in order to make it a priority to use TR when appropriate. In the context of the pandemic, the requirements with respect to post-stroke rehabilitation follow-ups were also adjusted to ensure that continuity of services was maintained. Moreover, the objective to offer at least a minimal amount of services to as many post-stroke individuals as possible led to a greater tolerance for offering less frequent, less varied or less intense services in comparison to the pre-COVID-19 period. This context allowed clinicians to develop experience using TR and thus better assess its strengths and weaknesses. For example, the finding that the therapeutic relationship can be developed in TR in a similar way to face-to-face follow-up echoes the results of other studies in the field of telehealth [30–32] and can be considered a facilitator for the implementation of TR. In addition, the fact that clinicians have opted to carry out specific clinical activities, such as assessments or interventions that are standardized or involve a safety risk in person despite the context of the pandemic, shows that there were limits to TR for clinicians. ESD clinicians exhibited flexibility in their clinical activities and interdisciplinary work to offer quality services that they perceived to be optimal by collaborating to make the few face-to-face visits to clients' homes as profitable as possible. Similarly, interdisciplinarity was also a facilitator in a case study where TR was used for stroke rehabilitation [33]. The particularities of the ESD program, the services offered there and the clientele served illustrate, however, that it is difficult to provide services entirely by TR, which confirms the relevance of a hybrid model where clinicians can choose between face-to-face or TR interventions in the usual offer of ESD services, which was also presented as promising in a recent scoping review on how clinicians used TR in stroke rehabilitation [34].

Thus, this study of TR implementation suggests that it is influenced by multiple factors related to the technology used, the clinicians who use it, the clinical team and the context in which they exercise their functions. Being able to compare pre- and during-COVID TR implementation and actual use highlighted the importance of taking into account familiarity with technology and its correspondence with clinical needs, as well as the need for adequate support from a technical team or clinical champions within the clinical team being critical to its use. Taking into account the elements influencing the implementation

of TR in usual practice therefore requires a multifactorial and dynamic analysis over time. The multifactorial nature of implementation of TR in stroke rehabilitation has previously been outlined in numerous studies [35] and continues to be further understood through TR use in real-life settings.

*Strengths and Limitations*

This study has methodological strengths that further the credibility of its results [36]. In fact, the prolonged contact of the first author with the ESD team during the pre-COVID-19 implementation period, the co-facilitation of each focus group, using the CFIR [22] to orient data collection and analysis and the regular discussions between the coder and the two facilitators of the focus groups to deepen the thematic analysis are all strategies that promote the trustworthiness of our analysis and reliability of what our participants shared. Moreover, the fact that most of the ESD program's stakeholders participated in the study promoted the diversity of points of view and fostered an analysis that is representative of stakeholders' perceptions in stroke rehabilitation programs that are alike.

This study also has certain limitations. The sample, which was of a relatively small size, remains representative of the clinical team in place. Given that participants were from the same ESD program, transferability of the results to other stroke rehabilitation programs using TR may be limited, especially inpatient and outpatient programs, which are less representative of the working context of participants reported here [36]. However, the methods and results were clearly described, facilitating the contextualization of our results to differing contexts. Furthermore, the structure of the interdisciplinary ESD team is similar to that found in the regional health care system. Qualitative study designs that include an in-depth analysis of the context of use, such as reported here, can contribute to a better understanding of usability, acceptability, challenges and satisfaction, as well as facilitating the implementation and adoption of TR in clinical practice [11,37]. By presenting an in-depth description of the context and actual use of TR in an ESD program, clinicians, managers, patients and researchers can better explore possible opportunities to include TR. As addressed by Eng and Pastva [38], stroke TR research is evolving. Many studies continue to take place in controlled research settings and it is therefore important to expand current research across the continuum of stroke rehabilitation, as well as examine its use in actual clinical practice.

Future studies should also examine and compare clinicians and managers' perspectives to those of post-stroke individuals and their caregivers, as well as examine clinical outcomes and cost-effectiveness associated with TR for ESD using larger scale programmatic efficacy and feasibility studies in order to further advance TR use and increase accessibility to optimal rehabilitation care throughout the continuum of care.

## 5. Conclusions

This study showed that implementing TR involves several interrelated factors, ranging from the technology used to the individuals who use it and the context in which it is implemented. The specific context of the ESD program and the post-stroke clientele it serves also raise the need for an adapted approach in the implementation of TR. Motivation to use TR, in the case of this study being in part related to COVID-19, has been shown to be a determining factor in the adoption of TR by ESD clinicians who saw an advantage for them and their clients in using it. Future studies of the implementation of TR in real post-stroke rehabilitation contexts should therefore provide for multiple and complementary strategies in order to promote their success and focus on factors that will highlight the usefulness and added value of TR for clinicians and for the quality of the services they provide to their clientele. Factors influencing implementation of TR in other rehabilitation programs and contexts, for individuals with stroke and for people with other health conditions should be assessed in future research work. Moreover, future studies addressing the implementation of TR in ESD stroke rehabilitation should compare patient outcomes for different models

of care which are emerging, such as using TR exclusively, using it as a hybrid model or providing fully in-person rehabilitation.

**Author Contributions:** Conceptualization, D.K. and A.R.; Methodology, D.K., L.-P.A. and A.R.; Validation, O.C., R.G. and A.R.; Formal Analysis, E.M., L.-P.A., A.R. and D.K.; Investigation, L.-P.A., E.M. and D.K.; Resources, R.G.; Data Curation, L.-P.A. and E.M.; Writing—Original Draft Preparation, L.-P.A. and E.M.; Writing—Review and Editing, D.K., A.R., O.C. and R.G.; Visualization, D.K. and R.G.; Supervision, D.K.; Project Administration, D.K. and L.-P.A.; Funding Acquisition, D.K., A.R., O.C. and R.G. All authors have read and agreed to the published version of the manuscript.

**Funding:** This research was funded by the Ministère de l'économie, des sciences et de l'innovation du Québec (Project #2-34). The authors gratefully acknowledge that the first author was supported by doctoral scholarships from the Canadian Institutes for Health Research, the Fonds de recherche du Québec en santé (FRQS), the School of Rehabilitation of the Université de Montréal (UdeM), the Canadian Occupational Therapy Foundation (COTF) and the Ordre des ergothérapeutes du Québec (OEQ). The last author was supported by a career award from the FRQS.

**Institutional Review Board Statement:** The study was conducted in accordance with the Declaration of Helsinki, and approved by the Research Ethics Board of the Centre for Interdisciplinary Research in Rehabilitation of Greater Montreal (CRIR) (protocol code CRIR-1347-0618 approved on 5 November 2018).

**Informed Consent Statement:** Informed consent was obtained from all subjects involved in the study.

**Data Availability Statement:** For reasons of confidentiality and in order to comply with the requirements of the research ethics committee, no research data is made available.

**Acknowledgments:** The research team would like to warmly thank the clinicians and managers who took part in this study.

**Conflicts of Interest:** The authors declare no conflict of interest.

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
