# Peer review of "Implementation of Telerehabilitation in an Early Supported Discharge Stroke Rehabilitation Program before and during COVID-19: An Exploration of Influencing Factors"

_disabilities, doi:10.3390/disabilities3010007_

Round 1

Reviewer 1 Report

I would like to thank the authors for their hard work. In my humble opinion, despite some limitations that the authors mention and discuss, this is an interesting work, that was responsibly conducted and reported. 

Author Response

Thank you for your comments.

Reviewer 2 Report

Please see file attachment. 

Reviewer 3 Report

Thank you for your very well written manuscript. It especially highlights the impact of Covid-19, and the opportunities in adversity that Covid-19 presented for digital health. The manuscript can do with some minor typographical editing and corrections without further review by me. 

Author Response

Thank you for your comments.

Round 2

Reviewer 2 Report

Thank you for your attention to the suggested revisions. A major question about the merit of this study lies in the limited number of participants. Thus the overall conclusions are speculative at best. A larger N would significantly improve the validity of this study. 
